# Empowering refugee voices: Using Nominal Group Technique (NGT) with a diverse refugee Patient Advisory Committee (PAC) to identify health and research priorities in Calgary, Canada

**Deyana Altahsh**[1,2], **Linda Holdbrook**[1], **Eric Norrie**[1,3,4], **Adanech Sahilie**[1], **Mohammad Yasir Essar**[1], **Rabina Grewal**[1,3], **Olha Horbach**[1], **Fawzia Abdaly**[1], **Maria Santana**[1,3,5], **Rachel Talavlikar**[1,6], **Michael Aucoin**[1,6], **Annalee Coakley**[1,6], **Gabriel E. Fabreau**[1,3,4]*

1 Refugee Health YYC, O'Brien Institute for Public Health, Cumming School of Medicine, University of Calgary, Calgary, Canada, 2 Faculty of Medicine & Dentistry, University of Alberta, Edmonton, Alberta, Canada, 3 Department of Community Health Sciences, Cumming School of Medicine, University of Calgary, Calgary, Alberta, Canada, 4 Department of Medicine, Cumming School of Medicine, University of Calgary, Calgary, Alberta, Canada, 5 Departments of Pediatrics, Cumming School of Medicine, University of Calgary, Calgary, Alberta, Canada, 6 Department of Family Medicine, Cumming School of Medicine, University of Calgary, Calgary, Alberta, Canada

* gefabrea@ucalgary.ca

## Abstract

### Background

Despite rising forced displacement globally, refugees' health and research priorities are largely unknown. We investigated whether a diverse refugee committee could utilize participatory methods to identify health priorities and a research agenda to address them.

### Methods

We conducted a qualitative study with focus groups of current and former refugees, asylum claimants and evacuees from a specialized refugee clinic over a year in Calgary, Alberta, Canada. We collected sociodemographic data using standardized instruments, then utilized a four-step nominal group technique process (idea generation, recording, discussion, and voting) to identify and rank participants' health and research priorities. Participants ranked their top five priorities across three time periods: Pre-migration/early arrival (0–3 months), post-migration (3 months–2 years), and long-term health (>2 years). Participants created overarching priorities and corroborated findings via a member checking step.

**Data availability statement:** All relevant data (de-identified only) are within the paper and its Supporting Information files.

**Funding:** The SPECIAL CATALYST COMPETITION to support: A Systems Approach to Optimizing Resources for Vulnerable Populations: Data Management and Collaborative Action, awarded by the O'Brien Institute for Public Health Research Catalyst Grant, was received by GEF. This funding was awarded as part of the October 2022 Competition (no specific grant number) and supported the development of the study. Funding was provided by the O'Brien Institute for Public Health, University of Calgary. Further information on this grant can be found on the funder's website: https://obrieniph.ucalgary.ca/catalyst-funding-catalyst-projects The sponsors or funders played no role in the study design, data collection and analysis, decision to publish, or preparation of the manuscript.

**Competing interests:** The authors have declared that no competing interests exist.

## Findings

Twenty-three participants (median age 35 years) attended one or more of five focus groups. Twenty-one completed sociodemographic surveys: 16/21 (76%) were women, representing 8 countries of origin. Participants identified "more family physicians" and "improving health system navigation" (11/60 votes each) as top health and research priorities respectively across all resettlement periods. Participants also prioritized pre-departure healthcare system orientation and improved post-arrival and long-term mental health services. Twelve participants completed the member checking process, affirming the results with minor clarifications.

## Interpretation

This proof-of-concept study illustrates how refugees can use a rigorous consensus process without external influence to prioritize their healthcare needs, direct a health research agenda to address those needs, and co-produce research. These low-cost participatory methods should be replicated elsewhere.

## Introduction

The number of forcibly displaced people globally has reached unprecedented levels, now estimated at 117.3 million people [1]. Historically, Canada is a refugee resettlement leader, with Toronto and Calgary currently being its Eastern and Western resettlement hubs [2]. Newly arrived refugees face high social vulnerability that negatively impacts their health and long-term well-being including, discrimination, health system navigation difficulties, language barriers, poverty, and employment challenges [3,4]. Refugees also face greater access barriers to healthcare and education, have less family support, and fewer financial resources than host populations, that combined, negatively impact their post-resettlement health and wellbeing [3–5].

To address these health disparities, many jurisdictions have developed specialized refugee health clinics in Canada to serve refugees' and migrants' unique health and resettlement needs [2]. Despite these specialized refugee clinics, gaps remain in providing refugee-centered healthcare services. Community-based participatory research (CBPR) techniques, such as Delphi and Nominal Group Technique (NGT), have been used to identify community-driven healthcare priorities and encourage democratic participation of socially marginalized groups [6,7].

Despite these advances, refugee and asylum seeker voices are often unrepresented in traditional health research [8] leading to major knowledge gaps about their perceived healthcare and research priorities during pre, early-post, and late-post resettlement periods. Engaging directly with resettled refugees can address these gaps and inform health system improvements; however, whether refugee can utilize these techniques to derive their own health priorities, and a research agenda to address them, remains untested. For example, the WHO recently released an important Global Research Agenda on Health and Migration[9], which details a globally derived consensus on the fields' research priorities, but notably omitted refugees [9].

To address these critical gaps in inclusive health and research agenda-setting among forcibly displaced populations, we employed CBPR techniques to explore consensus-derived health and research priorities among a diverse committee of resettled refugees, asylum claimants and evacuees. We aimed to investigate the feasibility and effectiveness of employing these techniques with multilingual and multicultural displaced communities to identify health and research priorities across different phases of resettlement.

## Materials & methods

### Study design and setting

We conducted a qualitative study that utilized pragmatist epistemology and NGT among refugees, asylum seekers and evacuees in Calgary, Canada from January 2023 to May 2024. We recruited a Patient Advisory Committee (PAC) of current and former refugees, asylum claimants and Ukrainian evacuees (herein referred to as 'refugees') from a specialized multidisciplinary refugee clinic and various partner community organizations [10]. We aimed to investigate healthcare delivery gaps and priorities among refugees, and research priorities aimed to address them. We conducted evening focus groups in person at the refugee clinic, with online video meetings for participants unable to attend in-person. We assessed NGT process feasibility among PAC participants using participation rates, session retention, and successful completion of the priority-ranking process. We evaluated NGT process effectiveness depending on whether PAC participants generated actionable health and research priorities and whether participants validated these priorities during a final member checking step.

### Inclusion, recruitment and sampling framework

We included adult refugees (18+ yrs) living in Canada who could share their healthcare system experiences and communicate with basic English proficiency. Clinic staff and non-study physicians informed and referred potential participants to the study team. Recruitment occurred from January 5th, 2023, until January 29th, 2024. Potential committee members were purposively recruited to include various global regions, religions, cultures, languages, gender identities and refugee categories (i.e., government-assisted, privately sponsored or refugee claimant). Community scholars (community leaders and research team members) also facilitated participant recruitment [11].

We later included Ukrainian evacuees despite not being classified as refugees by the Canadian government given they represented a large forcibly displaced population with similar needs, and experiences as other refugees [12]. We adapted our protocol to include evacuees from the Canada Ukraine Authorization for Emergency Travel (CUAET) program in March 2022 [13] as Russia's invasion of Ukraine occurred before the study began in February 2022. We first hosted one focus group with Ukrainian evacuees to better understand their health and research priorities. Observing parallels in responses with the existing refugee PAC, we subsequently merged the groups. Our full protocol is available as supplementary material for further reference.

### Data collection

At each focus group, participants were given time to ask questions or request translations to provide informed consent. Food and refreshments and $25 gift card per person were provided for each focus group attended. Participants completed a sociodemographic survey form using standardized questions adapted from the Canadian Community Health Survey [14]. We collected self-reported age, sex at birth, gender, country of origin, ethnicity, languages spoken, education level, refugee category, family composition, income, and relevant migration histories. Participants completed paper or online surveys via Qualtrics (Provo, Utah, United States). All participants in this study provided informed written consent prior to participation. All data was securely stored at the University of Calgary.

## Study time periods

Participants identified health and research priorities across three time periods: Pre-migration to early arrival (-3–3 months), post-migration (3 months–2 years), and long-term resettlement (>2 years). We defined pre-migration to early arrival as 3 months pre-departure to 3 months post-arrival to align with United States resettlement policy that expects economic independence after 90 days of arrival [15]. We defined the post-migration period as 3 months to 2-years post-arrival as most specialized Canadian refugee clinics provide care for newly arrived refugees for 1–2 years before transitioning to the general health system [2]. We defined long-term resettlement as >2 years post arrival.

## NGT process and focus groups

We conducted focus groups utilizing NGT [6,7,16], a qualitative consensus-building methodology, which captures all perspectives inclusively and democratically within a group setting to generate participant priorities and build consensus [6]. The NGT format included:

1) Generating Ideas: Facilitated idea generation by each participant silently.

2) Recording Ideas: Participants shared ideas one at a time, without interruptions or discussion to allow total participation. A facilitator recorded all responses on a whiteboard verbatim, creating numbered lists for research and health priorities. During this step, discussion or questions were not permitted to ensure free expression by all.

3) Discussing Ideas: Once all ideas were recorded, participants openly discussed each for clarity, to ask questions, or to agree or disagree.

4) Voting on Ideas: Participants anonymously selected and ranked their top 5 ideas from most important to least important on cue cards and submitted them to the study team, who aggregated vote points. Ideas with the most vote points represented the group consensus.

Three facilitators led PAC meetings: a primary facilitator for meeting structure, one to record field notes; and a third to record ideas and priorities on physical and virtual whiteboards. Study facilitators were study team members and study co-authors. We did not record initial focus groups to ensure confidentiality and maintain participants' comfort, opting instead to take detailed field notes, including verbatim participant quotes. We recorded the final group discussion given its large size to ensure all points were captured. Occasionally, similar ideas were noted separately and subsequently voted upon. To address this redundancy, two authors independently reviewed the resultant priorities to identify and merge similar concepts and their corresponding point allocations. Any remaining conflicts were resolved through consultation with a third senior author and consensus reached through group discussion.

## Priority ranking and analysis

We used dense ranking to analyse and present lists of participants-derived priorities to emphasize categorization over a rigid hierarchy. After participants ranked their priorities, they were scored from 5 points for the first priority to 1 point for the fifth. The maximum achievable score for each priority was calculated by multiplying the total number of participants by 5, representing the highest possible score if every participant selected a specific priority as their top option [6]. This score represented the denominator to which the total votes were compared against and ranked. Sociodemographic data was analyzed using Excel Software (Redmond, Washington, United States). We calculated means and standard deviations for normally distributed data, medians with interquartile ranges for non-normally distributed data, and frequencies for categorical data.

### Overarching priorities

Once priorities were identified across the three resettlement periods, participants convened for a final focus group to determine the overarching priorities in both health and research domains across all three time periods. Each participant received five stickers to represent their health priorities and five for their research priorities and instructed to distribute stickers freely across the previously identified priorities list across three different resettlement periods. They could allocate multiple stickers to a single priority if desired. Finally, we calculating the total cumulative sticker counts to identify the top five overarching health and research priorities across resettlement time periods.

### Member-checking

After all focus groups, we conducted a member-checking step utilizing an online survey to verify results with PAC participants [17]. In the member-checking process researchers share findings with participants to verify their accuracy and ensure credibility. In keeping with participatory research principles, it also invites participants to actively engage in the analytic process, thus enhancing the study finding's accuracy, validity and acceptability [17]. We sent participants consolidated tables of the identified priorities for feedback and confirmation, and asked participants to opine on the study's key findings, and their implications for Canadian refugee healthcare. We incorporated this member-checking feedback into the results. Finally, participants were invited to review the study manuscript, suggest edits, and co-author if interested.

### Quotes

The first author conducted a thematic analysis of quotes taken verbatim from either field notes or feedback obtained during the member checking step. The analysis was subsequently reviewed and finalized by two other co-authors.

This study was approved by the Calgary Conjoint Health Research Ethics Board (CHREB) (Ethics ID: REB21–0954_REN2). We utilized the Guidance for Reporting Involvement of Patients and the Public (GRIPP2) and Consolidated Criteria for Reporting Qualitative Research (COREQ) checklists to guide and summarize the study [18,19].

## Results

We recruited 23 focus group participants among which 21 individuals completed sociodemographic surveys. Table 1 presents their characteristics. Overall, PAC participants were diverse, representing 8 distinct countries and different resettlement durations in Canada ranging from 9 months to 18 years of residence (Table 1). Participants' households varied from 1 person to 8 people, and reported speaking 7 primary languages at home, however education level varied minimally, as all had completed some or full college or university education (Table 1). Notably, 29% of participants were naturalized citizens, and 24% of participants reporting mostly speaking English at home despite not having English as their first language. All temporary foreign workers were Ukrainian evacuees.

Twelve participants completed the member-checking process, and all affirmed the results; only one participant requested clarification. Member-checking feedback was incorporated into the results (Figs 1, 2, 3, 4, and 5).

### Pre-migration/early arrival (0–3 months)

In the pre-migration/early arrival period, participants identified healthcare system navigation as the top priority and the greatest health need. Participants highlighted including the need to increase basic Canadian healthcare system information and understanding, eligibility and access to medications (especially for pre-existing medical conditions), and knowing if pre-conditions impart ineligibility for refugees to come to Canada. Similarly, health system navigation post-arrival secured a unanimous vote as the top research priority (55/55) with Canadian health system navigation pre-arrival as the second research priority (42/55) (Fig 1).

**Table 1. Qualitative participant demographic information.**

| Characteristics | Number of Participants (n=21) (%) |
|---|---|
| **Sex & Gender** | |
| Cismen | 5 (23·8) |
| Ciswomen | 16 (76·2) |
| **Age** | |
| Median (years) [IQR]† | 35 [26-46·5] |
| **Cultural Group*** | |
| Arab | 2 (9·5) |
| Black African | 5 (23·8) |
| South Asian | 1 (4·8) |
| Ukrainian | 1 (4·8) |
| West Asian | 4 (19·0) |
| White European | 8 (38·1) |
| **Country of Origin*** | |
| Afghanistan | 3 (14·3) |
| Cameroon | 1 (4·8) |
| Eritrea | 2 (9·5) |
| Ethiopia | 1 (4·8) |
| Iraq | 1 (4·8) |
| Missing | 2 (9·5) |
| South Sudan | 1 (4·8) |
| Syria | 2 (9·5) |
| Ukraine | 8 (38·1) |
| **Years in Canada** | |
| Median years [IQR]† | 1.5 [1–4] |
| **Number of Persons Living in Household** | |
| Median [IQR]† | 3 [2–4] |
| **Languages Most Often Spoken at home*** | |
| Amharic | 2 (9·5) |
| Arabic | 1 (4·8) |
| Dari | 4 (19·0) |
| English | 5 (23·8) |
| Persian | 1 (4·8) |
| Tigrinya | 2 (9·5) |
| Ukrainian | 8 (38·1) |
| **Highest Education Completed** | |
| Completed college/university | 17 (81·0) |
| Some college/university | 4 (19·0) |
| **Immigration Status*** | |
| Government Assisted Refugees | 4 (19·0) |
| Landed Immigrant/Permanent Resident | 3 (14·3) |
| Naturalized Canadian Citizens | 6 (28·6) |
| Temporary Foreign Worker | 8 (38·1) |
| **Household Income Range** (in thousands of Canadian Dollars) | |
| Less than 10k | 4 (19·0) |
| Less than 25k | 1 (4·8) |

*(Continued)*

**Table 1.** (Continued)

| Characteristics | Number of Participants (n=21) (%) |
|---|---|
| less than 40k | 6 (28·6) |
| less than 50k | 3 (14·3) |
| less than 75k | 2 (9·5) |
| less than 100k | 1 (4·8) |
| more than 100k | 1 (4·8) |
| Missing | 3 (14·3) |

† IQR= Interquartile Range, *presented in alphabetical order

Post-arrival healthcare system navigation was also identified as a priority. Participants proposed potential solutions to address this gap, such as leveraging existing language classes to increase health system navigation post arrival, including connection to refugee-serving primary care clinics, vaccinations, insurance coverage, and the immigration medical exam, as well as where to start and the importance of each.

The Ukrainian focus group similarly identified healthcare navigation as the top priority (32/35, Fig 2) with their focus paralleling the main focus group. However, Ukrainian participants focused more on addressing the healthcare navigation barriers and increasing accessibility to available services. Full details on the verbatim and summarized research priorities voted on presented in Fig 1 are found in Appendix S2b and S2c Tables respectively.

### Post-migration (3 months-2 years)

In this time period, participants unanimously voted mental health as both the top research priority (25/25), emphasizing further need for research on this topic, and health priority (17/20), highlighting its importance in the middle settlement period (Fig 3). Participants identified health system navigation as their fourth health priority, highlighting its importance across settlement time periods (Fig 3). Participants indicated that mainstream approaches for health orientation did not work sufficiently in refugee communities, emphasizing that health information and updates were usually acquired informally through social media groups and community leaders. Thus, they recommended that governments and health systems better utilize social media to disseminate reliable health information. Participants provided other potential solutions, such as texting refugees in their language instead of calling in English, and using WhatsApp instead of emails, as preferred methods for communication.

*"Most of the information we get about Canadian healthcare we get from Facebook and Instagram"*

Various participants emphasized that social media is the primary platform for transmitting health information within the refugee community and reinforced their recommendations for healthcare information to be disseminated to refugee communities through these channels.

### Long-term resilient healthcare system (beyond 2 years)

As displayed in Fig 4, participants identified primary care physician shortages as a major Canadian healthcare system gap. They ranked International Medical Graduates' (IMG) support as a top priority in both health and research, as a solution to addressing this gap and sustaining an adaptable and resilient healthcare system for refugees.

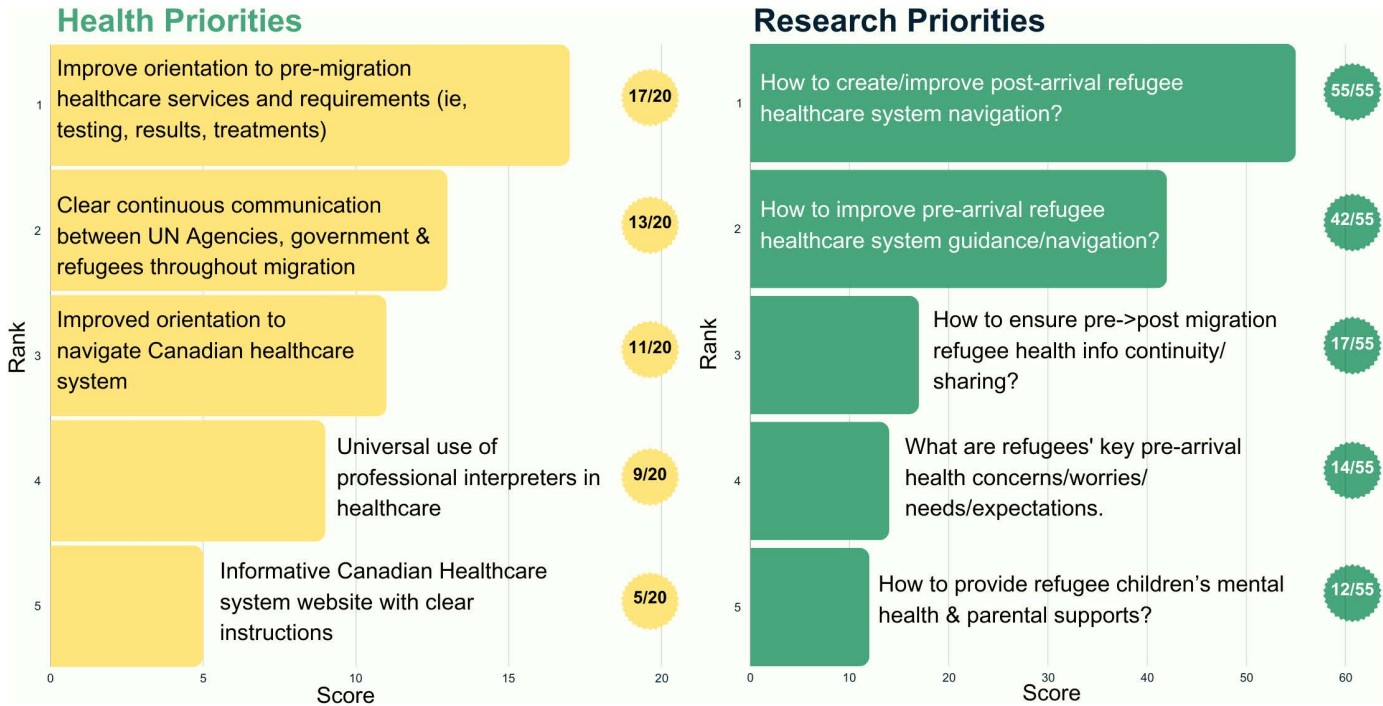

**Fig 1. Top five health and research priorities voted on for the pre-migration/early arrival time period (0-3 months).**

## Overarching priorities across all three time periods

We present the top five overarching priorities across resettlement periods in Fig 5. The committee identified improving the family physician supply as the top health priority and investigating ways to improve pre-migration health system orientation and navigation as the top research priority. The second major theme the committee identified was difficulties navigating the healthcare system. As seen in Fig 5, healthcare navigation made the 3rd health priority (8/60) and the 1st research priority (11/60).

> *"Even me as a medical doctor, don't know how to navigate healthcare system here in Canada…"*

Navigating the healthcare system is crucial for care access and allowing refugees to address the issues and conditions they are facing.

> *"It's so ridiculous that some families are going back to dangerous cities in Ukraine because they don't know how to navigate healthcare system here".*

Therefore, improving healthcare system navigation, was identified as critical health and research priorities for refugees across all time periods.

Many participants discussed experiencing difficulties adjusting to life in Canada, resulting in reduced mental health. As such, mental health was the third priority for both the health and research (7/60, 5/60 respectively, Fig 5), emphasizing its importance across all resettlement time periods.

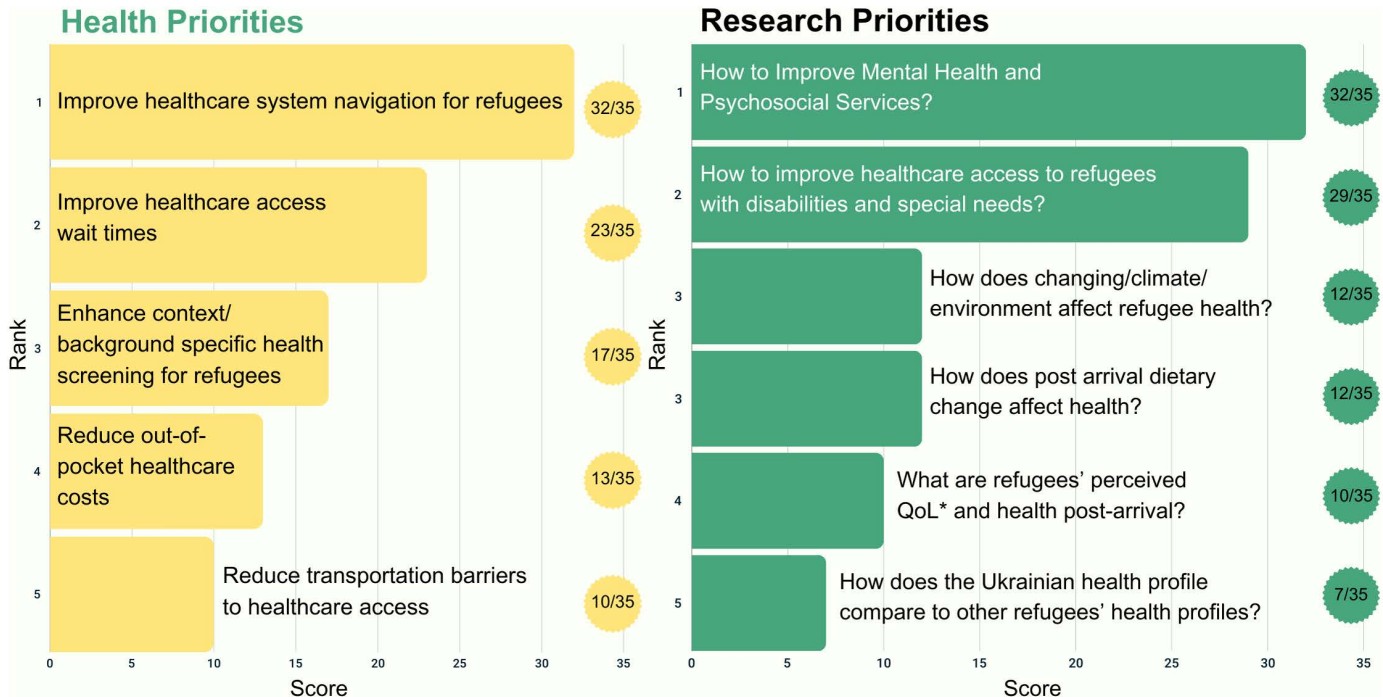

**Fig 2. Top five health and research priorities voted on for the pre-migration/early arrival time period (0-3 months) among Ukrainian Evacuees.**
*QoL: Quality of life.

Finally, in our member checking process, participants' key messages illustrated their call for increased representation and action to address identified gaps in healthcare for refugees.

*"If some services are not clear or available, it does not mean that you should give up…if we express a need and interest, over time the system will begin to see this need and respond to it. Our opinion is important."*

*"Canada accepts many refugees, many of them remain in the future to live in Canada. The more effectively the system takes care of health and availability of services at the beginning, the healthier residents we will have."*

## Discussion

This study demonstrates that a group of resettled multi-national, multi-lingual forcibly displaced refugees, asylum claimants and evacuees can utilize rigorous participatory research methodology to develop their own healthcare and health research priorities. This proof-of-concept study illustrates that despite various sociocultural differences, the forcibly displaced can effectively use a structured, low-cost, democratic consensus process to co-produce research, prioritize their healthcare needs, and shape research agendas to address these needs across different resettlement time periods. The NGT process was feasible, as demonstrated by high participation rates, strong session retention, and successful completion of the priority-ranking exercise. It was also effective: refugee participants identified actionable health and research priorities, which were subsequently validated during the member-checking step, confirming their relevance and applicability. To our knowledge, this is the first study to detail how refugees can set their own health research priorities (without experts' involvement and restricted area of focus) and co-produce research, particularly in the North American context. Notably,

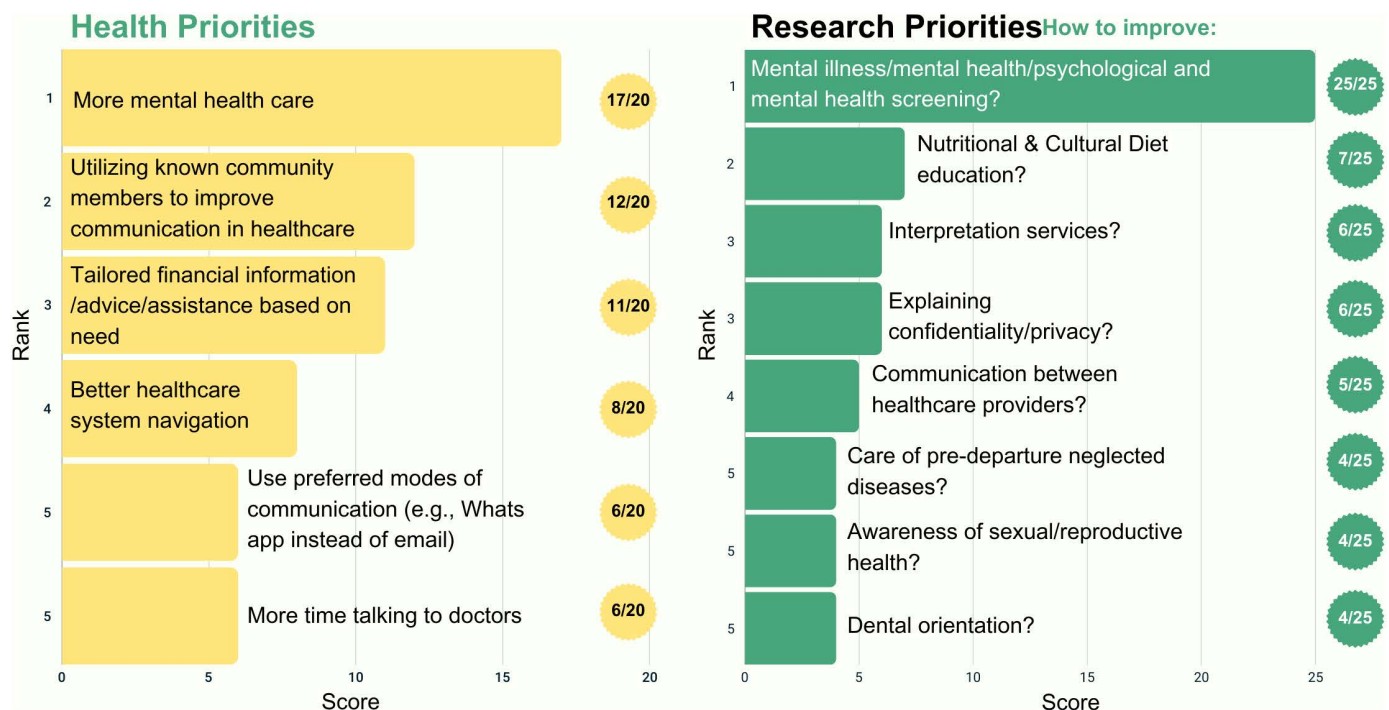

**Fig 3. Top five health and research priorities voted on for the post-migration time period (3 months – 2 years).**

across early, mid and long-term resettlement periods, improving health system navigation (pre- and post-arrival) and addressing refugee mental health emerged as critical overarching priorities, highlighting key focus areas for future healthcare research and interventions. Also, nearly a third of our participants were naturalized Canadian citizens; thus, providing valuable insights into health and research priorities for long-term post-resettlement refugee health and wellbeing.

To our knowledge, this is the first study to openly deploy CBPR methods with refugees without a specific illness focus or external expert involvement, to identify and set their own health research priorities [20]. Our findings extend evidence from a recent study in New York that used Delphi methods to set health and research priorities among diverse community members [16]. While this study used similar methods [16], we are first to apply them with refugees, providing a comprehensive look into their self-prioritized health and research needs. The recently published WHO global research agenda [9], calls for affirmation of the right to participate in decision making, and the validity in considering lived experience as a form of expertise; but unfortunately, did not include refugees [9]. This proof-of-concept study provides important evidence that responds to the WHO's call and others to engage people with lived experience and give them voice to solve - what they consider - their own health issues [9,21].

Previous studies have highlighted the mental health needs and healthcare navigation barriers faced by resettled refugees [22,23]; however, none specifically investigated whether refugees would benefit from host country's healthcare system orientation in the pre-departure setting. Our study participants emphasized a critical need for pre-migration Canadian healthcare system orientation, including information about medical expenses, health insurance coverage, treatment continuity for existing health conditions, and whether health conditions can affect migration status negatively. These findings highlighted a critical gap and potentially simple solution that could easily be addressed with standardized, language and culturally adapted materials in the pre-departure period to provide needed health system orientation.

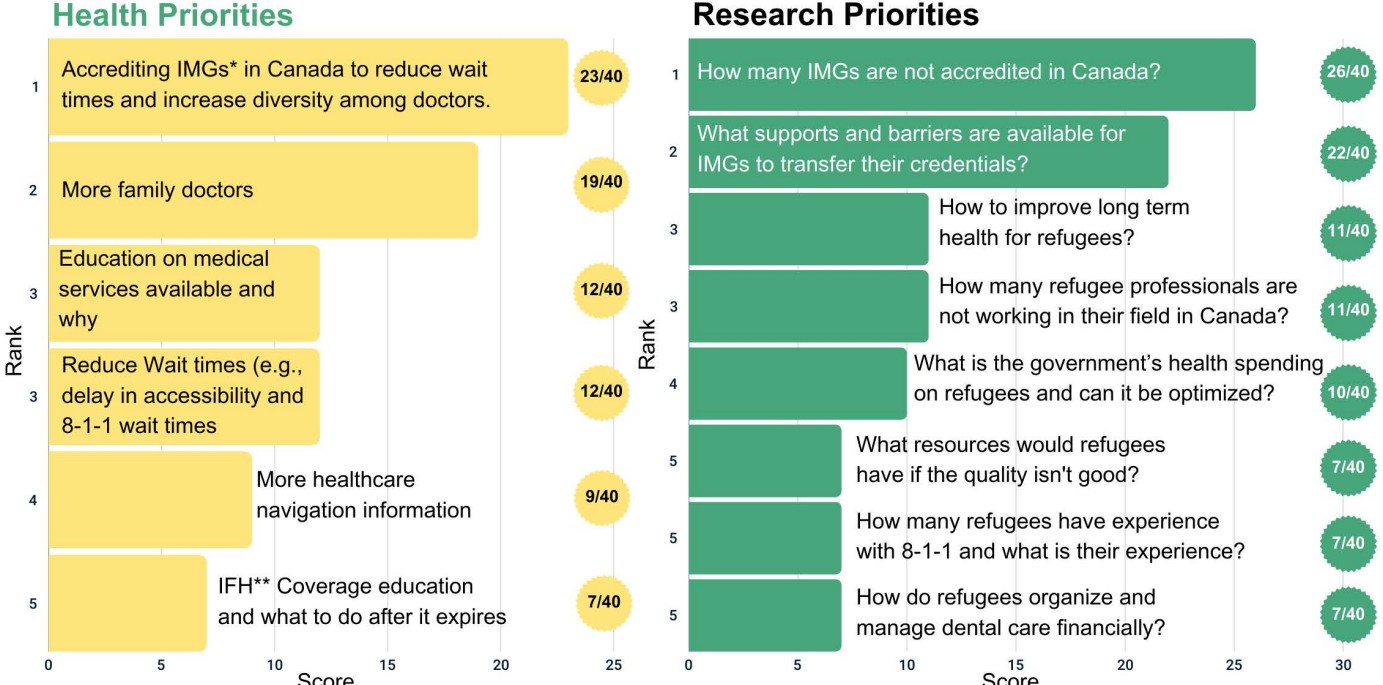

**Fig 4. Top five health and research priorities voted on for the long-term resilient healthcare system time period (beyond 2 years).** *IMGs: International Medical Graduates. **IFHP: Interim Federal Health Profram.

Our study identified other post-arrival healthcare navigation barriers such as, clinic and physician accessibility, confusing referral processes, lack of vaccination access, and difficulties with health insurance coverage. As in previous studies, these barriers seem especially evident for women refugees, who face increased healthcare barriers [24]. Our study participants, primarily women, identified many of the same barriers including managing lengthy wait times to access care, healthcare provider shortages, cultural and linguistic barriers, out of pocket costs, healthcare for children or elders, and difficult administrative processes [3,25]. Unlike previous studies, our participants developed nuanced healthcare and research priorities to address these barriers according to different resettlement periods.

Previous studies have shown that refugees' health status significantly deteriorates globally with longer residence in resettlement countries linked to greater declines in physical health [4,26]. These post-arrival health disparities are potentially related to unrecognized difficulties with health system access and navigation unaddressed early post-arrival, exacerbating access barriers thus over time [5]. Addressing these gaps with simple solutions offered by our PAC participants could help reduce the observed health deterioration refugees experience.

Language barriers among refugees may exacerbate difficulties with access and orientation to healthcare services [25]. Interestingly, although our study participants were highly educated and proficient in English, they still encountered healthcare navigation difficulties. This suggests that refugee groups with lower language proficiency or educational attainment may face even greater challenges [25], underscoring the need for universal professional translation services embedded within healthcare systems [27].

Improving mental health supports was a highly ranked overarching health and research priority, despite prevalent stigmas regarding mental health among many refugee communities [22]. Our participants priorities aligned with mental health and psychosocial supports as critically important recognized needs for forcibly displaced communities worldwide [3,28]. Limited access to mental healthcare can exacerbate other health issues, diminishing overall health and quality of life [29],

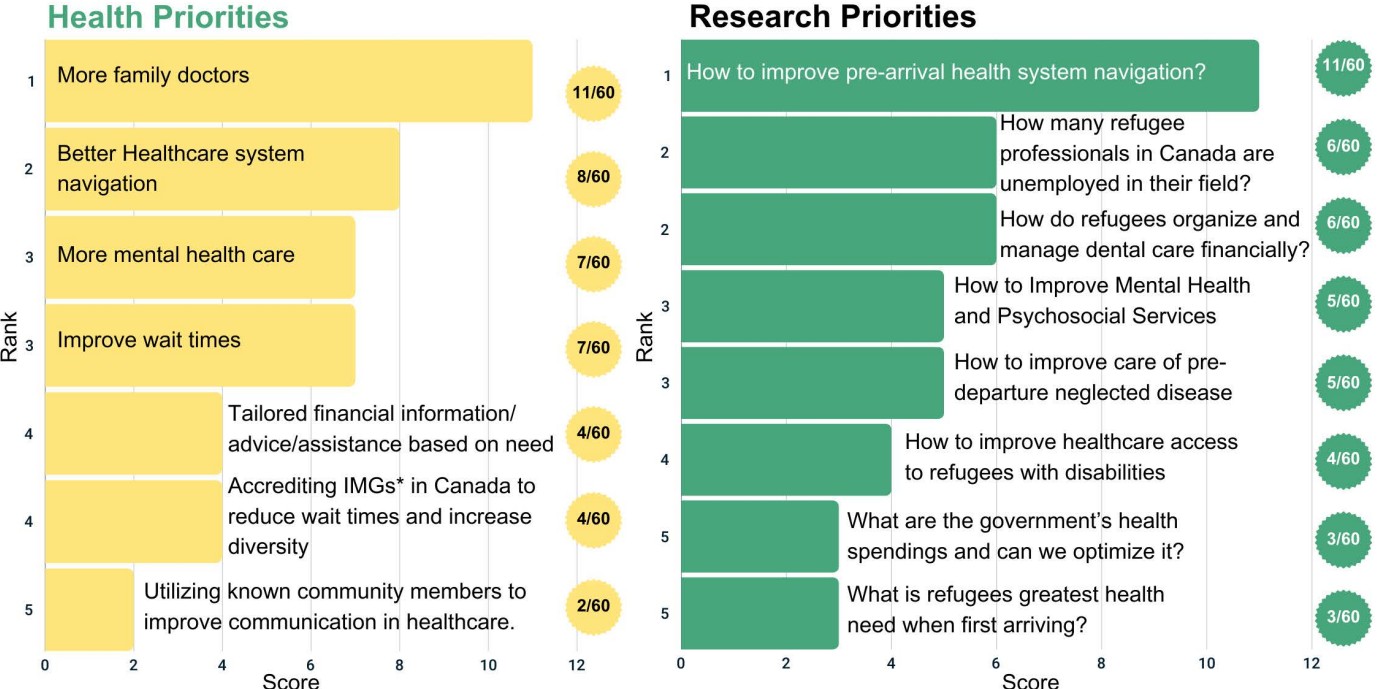

**Fig 5. Top 5 health and research priorities voted on across all time periods.**

highlighting the need for improved methods and research focused on increasing the accessibility and reach of culturally tailored mental healthcare and psychosocial supports for forcibly displaced populations [23]. Consistent with our results, a study conducted in U.S among Somali refugees discussed mental health needs and healthcare system navigation difficulties as their main themes [23]. Similarly, their cohort also suggested the use of community health workers to reduce system navigation difficulties and foster culturally sensitive engagement with the refugee community [23]. Similar studies focusing on mental health equity have proven the efficacy of including refugee community advisory boards to achieve mental health equity and reduce health disparities by fostering cultural humility, authentic engagement and respect for community norms and preferences [20].

Our study has limitations. First, our committee's limited educational diversity may constrain our findings' transferability, as our participants relatively high educational attainment and English proficiency is not representative of all forcibly displaced populations. Also, using English as a common language to conduct focus groups likely excluded other potential participants. Despite this, our participants' cultural, linguistic, geographical, and immigration categories were broadly diverse; thus, highlighting the applicability of the study's methods across diverse populations. Second, voluntary participation and anonymity of votes and responses may have led to incomplete data collection. For example, cue cards containing votes were sometimes incomplete or illegible and remote participants experienced infrequent disconnections, resulting in missed responses. These technical limitations were mitigated with member checking steps and participant inclusion throughout the study. Future work could utilize text to speech software, AI-facilitated real-time translations, or other technological solutions to avoid missing responses. Third, the real-time ideas generation documented verbatim by a note taker may have incompletely captured participants' ideas. However, open discussion at each focus group and member checking steps allowed participants to add nuance or clarify the priorities identified.

## Conclusions

This proof-of-concept study demonstrates that involving resettled displaced populations in health and research priority setting is feasible and effective, enabling refugees to set their own priorities and propose innovative solutions to existing healthcare barriers. These findings can guide health system leaders and policymakers to develop low-cost strategies that create more effective and inclusive healthcare systems and services that respond to refugees' lived experiences across different resettlement periods. Future research should explore these methods across other jurisdictions among forcibly displaced populations to help create more inclusive global healthcare, research, and policy agendas. Globally as forced displacement worsens and is expected to continue due to climate change, incorporating refugees' voices through participatory research methods is vital to creating refugee-responsive health systems and achieving the WHO's and United Nations' 2030 sustainable development goal of Universal Health Coverage [30].

## Supporting information

**S1 Table. Dates of focus group meetings and their topic of discussion.**
(DOCX)

**S2a Table. Raw rankings of research priorities for pre-migration/early arrival time-period.**
(DOCX)

**S2b Table. Grouped priorities with similar ideas and their combined total votes.**
(DOCX)

**S2c Table. Summarized similar priorities into single sentences with combined votes.**
(DOCX)\

**S2d Table. Final concise one sentence summary priorities for research priorities in pre-migration/early arrival time-period (0–3 months).**
(DOCX)

**S3 Study protocol. PAC Advisory Committee protocol.**
(DOCX)

**S4 Study data. PAC De-Identified study data.**
(XLSX)

## Acknowledgments

We thank the Calgary Refugee Health Clinic for providing us with the space to conduct our focus groups and the institutional support provided by the O'Brien Institute for Public Health at the University of Calgary Cumming School of Medicine. In particular, we thank our refugee patient advisory committee and the broader refugee community for their support, participation, and great input as the original impetus for this study. We value your time and your trust and hope that this work can inform the desired changes for newcomers.

## Author contributions

**Conceptualization:** Deyana Altahsh, Linda Holdbrook, Gabriel E. Fabreau.

**Data curation:** Deyana Altahsh, Eric Norrie, Adanech Sahilie, Rabina Grewal, Olha Horbach, Fawzia Abdaly.

**Formal analysis:** Deyana Altahsh, Eric Norrie, Gabriel E. Fabreau.

**Funding acquisition:** Linda Holdbrook, Gabriel E. Fabreau.

**Investigation:** Gabriel E. Fabreau.

**Methodology:** Deyana Altahsh, Linda Holdbrook, Eric Norrie, Adanech Sahilie, Maria Santana, Annalee Coakley, Gabriel E. Fabreau.

**Project administration:** Gabriel E. Fabreau.

**Resources:** Adanech Sahilie, Rachel Talavlikar, Michael Aucoin, Annalee Coakley, Gabriel E. Fabreau.

**Software:** Eric Norrie.

**Supervision:** Gabriel E. Fabreau.

**Validation:** Gabriel E. Fabreau.

**Visualization:** Deyana Altahsh, Rabina Grewal.

**Writing – original draft:** Deyana Altahsh, Gabriel E. Fabreau.

**Writing – review & editing:** Linda Holdbrook, Eric Norrie, Adanech Sahilie, Mohammad Yasir Essar, Olha Horbach, Fawzia Abdaly, Maria Santana, Rachel Talavlikar, Michael Aucoin, Annalee Coakley, Gabriel E. Fabreau.

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
