## [Decision Letter · Decision Letter 0]

3 Feb 2025

Dear Dr. Fabreau,

Thank you for submitting your manuscript to PLOS ONE. After careful consideration, we feel that it has merit but does not fully meet PLOS ONE’s publication criteria as it currently stands. Therefore, we invite you to submit a revised version of the manuscript that addresses the points raised during the review process.

We look forward to receiving your revised manuscript.

Kind regards,

Johanna Pruller, Ph.D.

Associate Editor

PLOS ONE

Journal Requirements:

3.  In the online submission form, you indicated that “Data are available upon request from authors”

Additional Editor Comments:

Please note that we have only been able to secure a single reviewer to assess your manuscript. We are issuing a decision on your manuscript at this point to prevent further delays in the evaluation of your manuscript. Please be aware that the editor who handles your revised manuscript might find it necessary to invite additional reviewers to assess this work once the revised manuscript is submitted. However, we will aim to proceed on the basis of this single review if possible.

The reviewer has raised a number of concerns that need attention. Could you please revise the manuscript to carefully address the concerns raised?

Reviewers' comments:

Reviewer's Responses to Questions

**Comments to the Author**

1. Is the manuscript technically sound, and do the data support the conclusions?

Reviewer #1: Partly

2. Has the statistical analysis been performed appropriately and rigorously?

Reviewer #1: N/A

3. Have the authors made all data underlying the findings in their manuscript fully available?

Reviewer #1: No

4. Is the manuscript presented in an intelligible fashion and written in standard English?

Reviewer #1: No

Reviewer #1: Thank you for the opportunity to review this interesting manuscript, which aims to describe the use of NGT with a diverse refugee patient advisory committee. Given the high levels of global displacement and the need to hear the voices of refugees, this paper adds important insights to the literature on refugee health. Nevertheless, I believe that major revisions are needed to make this paper ready for publication in PLOS ONE.

The authors state that they 'aimed to investigate the feasibility and effectiveness of employing these techniques [...]' (p5, l. 110), but the manuscript does not describe the investigation of these two aspects in any detail. I would have expected to learn more about how effectiveness and feasibility were investigated in the methods section and to see corresponding results in the results section. As this is not the case, the conclusion that NGT is effective and feasible seems to be a personal perception rather than the result of a thorough investigation.

The comparatively detailed description of materials and methods seems appropriate for a better understanding of the study setting and process, but some points need clarification:

- p5, l. 120: the month should not be abbreviated

- p5, l. 124: reference [10] refers to an article on cervical cancer screening - please check if this is the appropriate reference at this point and if so, please explain why in your response.

- p6, l. 139: Reference [11] refers to an article about COVID-19 outbreaks among migrant workers - the example of meat processing plants - please check if this is the appropriate reference at this point and if so, please explain why in your answer.

- p6, l.144: You mention that the study started before February 2022, on p.5, l.120 you state that you conducted the study between January 2023 and May 2024 - please clarify.

- - p6, l.144: please elaborate further on your protocol: is it publicly available or could be provided as supplementary material?

- p8, l. 181-192: Additional information is needed to better understand the NGT format and the respective roles and who is referred to as patient advisor, facilitator and participant. e.g.: What is the difference between patient advisor and participant?

- p10, l. 233-238: please provide further information on the member verification step to enable the reader to better understand who is considered a 'member'.

- p10, pg. 242: Who is referred to as a reviewer?

In addition, I see a need for further adjustments in other parts of the manuscript:

- Please elaborate further on data availability, as the PLOS ONE guidelines make clear: “Stating ‘data available on request from the author’ is not sufficient. If your data are only available upon request, select ‘No’ for the first question and explain your exceptional situation in the text box.”

- The title or at least the abstract should indicate the geographical entity in which the study was conducted.

- p1, l. 5: the affiliation of the first author should be clarified, a is not listed in l.9-22

- p12, table 1: the abbreviation IQR needs to be explained

- p16, l.321-322: the citation needs explanation/context

- p16, l.324-325: if this is a (sub)heading, it should not include the reference to figure 3

- please check that all references comply with the reference style

As the results of the study contain relevant findings for both healthcare and research, I would strongly encourage the authors to make the necessary revisions to publish these important results. I will be happy to review a revised manuscript.

**Do you want your identity to be public for this peer review?** For information about this choice, including consent withdrawal, please see our Privacy Policy

Reviewer #1: No

---

## [Author Response · Author response to Decision Letter 1]

2 Apr 2025

Dear Associate Editor Dr. Johanna Pruller,

We greatly appreciate the constructive feedback from the editor and reviewer, which have helped strengthen our manuscript entitled: Empowering Refugee Voices: Using Nominal Group Technique with a Diverse Refugee Patient Advisory Committee to Identify Health and Research Priorities in Calgary, Canada. We have attached our response to reviews, clean and tracked changes versions of our revised manuscripts as well as additional supplementary information including our study protocol and de-identified study data. We sincerely hope this addresses all the concerns and suggestions recommended.

Thank you again for your consideration in reviewing this revised submission as an original investigation to the PLOS ONE.

Sincerely,

Gabriel E. Fabreau MD, MPH, FRCPC

---

## [Decision Letter · Decision Letter 1]

15 Apr 2025

Empowering refugee voices: Using Nominal Group Technique (NGT) with a diverse refugee Patient Advisory Committee (PAC) to identify health and research priorities in Calgary, Canada

PONE-D-24-45096R1

Dear Dr. Fabreau,

We’re pleased to inform you that your manuscript has been judged scientifically suitable for publication and will be formally accepted for publication once it meets all outstanding technical requirements.

Kind regards,

Adetayo Olorunlana, Ph.D.

Academic Editor

PLOS ONE

Additional Editor Comments (optional):

Reviewers' comments:

Reviewer's Responses to Questions

**Comments to the Author**

Reviewer #1: All comments have been addressed

2. Is the manuscript technically sound, and do the data support the conclusions?

Reviewer #1: Yes

3. Has the statistical analysis been performed appropriately and rigorously?

Reviewer #1: N/A

4. Have the authors made all data underlying the findings in their manuscript fully available?

Reviewer #1: Yes

5. Is the manuscript presented in an intelligible fashion and written in standard English?

Reviewer #1: Yes

Reviewer #1: Dear authors, thank you very much for the thorough consideration of my comments. All my comments have been satisfactorily addressed and I have no further requests for revision. Keep up the good work.

**Do you want your identity to be public for this peer review?** For information about this choice, including consent withdrawal, please see our Privacy Policy

Reviewer #1: No

---

## [Editor Report · Acceptance letter]

PONE-D-24-45096R1

PLOS ONE

Dear Dr. Fabreau,

I'm pleased to inform you that your manuscript has been deemed suitable for publication in PLOS ONE. Congratulations! Your manuscript is now being handed over to our production team.

Kind regards,

on behalf of

Associate Professor Adetayo Olorunlana

Academic Editor

PLOS ONE